# Investigating the Catastrophic Forgetting in Multimodal Large Language Models

Yuexiang Zhai[1,*], Shengbang Tong[2], Xiao Li[3], Mu Cai[4],
Qing Qu[3], Yong Jae Lee[4,5], Yi Ma[1]

[1]UC Berkeley, [2]NYU, [3]University of Michigan, [4]University of Wisconsin–Madison, [5]Cruise LLC

Following the success of GPT4, there has been a surge in interest in multimodal large language model (MLLM) research. This line of research focuses on developing general-purpose LLMs through fine-tuning pre-trained LLMs and vision models. However, catastrophic forgetting, a notorious phenomenon where the fine-tuned model fails to retain similar performance compared to the pre-trained model, still remains an inherent problem in multimodal LLMs (MLLM). In this paper, we introduce EMT: **E**valuating **M**ul**T**imodality for evaluating the catastrophic forgetting in MLLMs, by treating each MLLM as an image classifier. We first apply EMT to evaluate several open-source fine-tuned MLLMs and we discover that almost all evaluated MLLMs fail to retain the same performance levels as their vision encoders on standard image classification tasks. Moreover, we continue fine-tuning LLaVA, an MLLM and utilize EMT to assess performance throughout the fine-tuning. Interestingly, our results suggest that early-stage fine-tuning on an image dataset improves performance across other image datasets, by enhancing the alignment of text and visual features. However, as fine-tuning proceeds, the MLLMs begin to hallucinate, resulting in a significant loss of generalizability, even when the image encoder remains frozen. Our results suggest that MLLMs have yet to demonstrate performance on par with their vision models on standard image classification tasks and the current MLLM fine-tuning procedure still has room for improvement.

## 1. Introduction

The recent progress in language models (LMs) has demonstrated impressive competency in engaging in a natural dialogue and in complex examinations [1–4]. Besides text generation, GPT4 [5] has recently shown impressive multimodal capability by performing a range of tasks with visual and language inputs. The emergent multimodal reasoning capabilities of GPT4 have propelled a surge of interest in multimodal large language models (MLLMs) [6–10]. This line of research typically involves (1) integrating pre-trained vision encoders [11, 12] with open-source LLMs [13–15], and (2) applying instruction tuning on the resulting vision-language models [7, 9, 10].

While many of these fine-tuned MLLMs have demonstrated remarkable capabilities in general purpose vision-language comprehension [16, 17], these models still suffer from *catastrophic forgetting* [18–21]. That is, the models tend to overfit to the fine-tuning dataset and consequently experience a decline in performance on pre-training tasks. Catastrophic forgetting in image classification has been extensively studied in computer vision and machine learning [22, 23]. However, recent developments in MLLMs [6–10] have mainly focused on creating multimodal chatbots for visual question answering [24], without evaluating their fundamental image classification capabilities, let alone explore the catastrophic forgetting in MLLM. That being said, prior MLLM evaluation frameworks [17, 25] mainly focus on assessing cognitive reasoning capability or hallucinations, which overlooks the necessity to critically examine how well MLLMs inherit the image classification capability from their base vision encoders [11, 12].

---

*Work done when YZ and MC were interning at Cruise LLC. Email: simonzhai@berkeley.edu. Project website: https://yx-s-z.github.io/emt/.

First Conference on Parsimony and Learning (CPAL 2024).

To comprehensively investigate the catastrophic forgetting in fine-tuned MLLM, we present the **E**valuating **M**ul**T**imodality (EMT) framework, which, to the best of our knowledge, is the *first* evaluation framework that studies the catastrophic forgetting in MLLMs. The EMT framework is a two-stage approach that treats each MLLM as an image classifier. In particular, for an input text and image pair, EMT first prompts the testing MLLM by asking it to classify the input image, and then post-processes the outputs to obtain a classification accuracy.

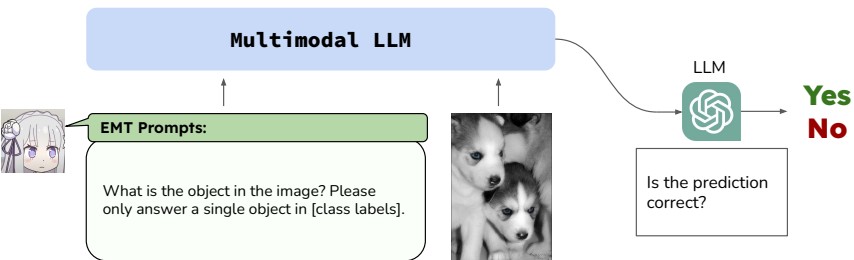

Figure 1: The EMT evaluation pipeline for MLLM. We prompt each MLLM as an image classifier by (1) inputting an image from a classification task; (2) asking the MLLM to explicitly answer a single label from the classification task. We evaluate the correctness of each output using another LLM.

We first apply EMT to several open-source fine-tuned MLLMs [7, 9, 10, 26] and observe a severe catastrophic forgetting phenomenon among *all* the tested models. That is, the majority of the tested MLLMs *fail to retain a comparable classification accuracy when compared to the zero-shot performance of their vision encoders*. After analyzing the results from the tested open-source models, we identify hallucination [8, 25, 27, 28] as one the major factors contributing to the performance degradation in MLLMs. Specifically, the tested MLLMs hallucinate by generating additional outputs that are irrelevant to the input question, including outputting more than one label or generating unverifiable descriptions of a label.

To gain deeper insights into how fine-tuning impacts the performance of MLLMs, we continue to fine-tune LLaVA [7], a popular MLLM achieving state-of-the-art accuracy on Science QA [29], and then apply the EMT evaluation to the fine-tuned LLaVA. Our fine-tuning experiments reveal two main observations. *Initially, fine-tuning on one dataset demonstrates generalization to other datasets*, as it improves the alignment between textual and visual features. However, *as the fine-tuning progresses, LLaVA starts to hallucinate* by disregarding the questions and exclusively generating text based on the examples in the fine-tuning datasets.

To summarize, this paper makes two key contributions.

- We propose EMT, an evaluation framework designed specifically to evaluate the phenomenon of catastrophic forgetting in MLLMs. To the best of our knowledge, EMT is the first evaluation framework to investigate catastrophic forgetting in MLLM through classification. Through EMT, we discover that nearly all the tested models fail to retain the classification performance of their vision encoders.

- We conduct fine-tuning experiments on LLaVA. Our fine-tuning results indicate that while moderate fine-tuning is advantageous for non-fine-tuned tasks, excessive fine-tuning ultimately leads to catastrophic forgetting in these tasks.

Our findings suggest that the fine-tuning process of MLLMs can still be further improved, particularly in mitigating catastrophic forgetting and reducing hallucinations.

## 2. Related Works

**Fine-Tuning and Catastrophic Forgetting.** Fine-tuning large pre-trained models has significantly transformed the field of natural language processing [1, 2, 30–32]. Despite its ubiquity and remarkable achievements, fine-tuning LLM still suffers from core machine learning problems such as catas-

trophic forgetting [33]. Catastrophic forgetting widely appears in LLM fine-tuning [19, 21, 34–36] or in-context learning [37, 38], as the LLMs tend to overfit to the small fine-tuning dataset resulting in losing performance on other tasks [34]. Various approaches have been proposed to mitigate the catastrophic forgetting problem in LLM fine-tuning, including pre-trained weight decay [36], learning rate decay [34], regularizations [35], and adversarial fine-tuning [19]. However, in MLLM, such a catastrophic forgetting phenomenon has not been thoroughly studied yet. Our work is most related to several evaluation metrics for MLLMs [17, 25], which proposed a comprehensive framework for evaluating the perception and recognition [17] or hallucinations [25], while the proposed EMT specifically aims at evaluating the catastrophic forgetting in MLLMs.

**Multimodal Large Language Models.** Multimodal Large Language Models (MLLMs) have emerged as a significant advancement in vision-language models, which significantly improves the model's reasoning capability. These models are designed to process and interpret information from multiple modalities, such as text and images, to perform complex tasks that require a comprehensive understanding of the context. Recent works [6–10, 26, 39–42] have contributed to the development and enhancement of MLLMs by leveraging the strong reasoning capability of LLMs such as LLaMA [14, 15]. LLaVA [7], as presented in the paper under discussion, represents a novel approach to instruction tuning on machine-generated multimodal language-image instruction-following data, achieving impressive multimodal chat abilities and state-of-the-art accuracy on Science QA [29]. Following the instruction tuning approach, various works came out focusing on other modalities such as video [43] and point cloud [44]. See Yin et al. [16] for a more comprehensive overview of the current state and future directions of MLLMs.

**A Theoretical Perspective of Catastrophic Forgetting through Minority Collapse.** Recently, Yang et al. [23] introduced an approach to address the issue of catastrophic forgetting, drawing inspiration from the principles of Neural Collapse (NC) [45–50]. In particular, Fang et al. [47] proposes *minority collapse* as a subsequent research direction of NC. Minority collapse describes a phenomenon in supervised learning with imbalanced data, where the classifiers of the minority classes converge to one vertex when the sample size ratio between the majority and minority classes reaches infinity. This result implies that all minority classes are indistinguishable when the imbalance ratio reaches infinity. To connect the fine-tuning with minority collapse: (1) Treating the absent class in fine-tuning as minority classes with a sample size of zero, directly implies the imbalanced training scenarios with a ratio of infinity; (2) Such an imbalance training in the fine-tuning phase will make the classifiers of the pre-trained classes converges to one vertex [47]; (3) Hence, the pre-trained classes become indistinguishable during fine-tuning, which results in catastrophic forgetting.

# 3. Fine-Tuning Image Classification

To verify the theoretical results inspired by minority collapse [47, 48], where supervised fine-tuning leads to catastrophic forgetting, we first perform pre-training and fine-tuning of image classification via ResNet [51]. Next, to further investigate the catastrophic forgetting in the vision-language model, we conduct experiments in fine-tuning the Contrastive Language-Image Pre-Training network (CLIP) [11].

## 3.1. Pre-Training and Fine-Tuning for Image Classification

To initiate our investigation, we train ResNet18 [51] on conventional image classification benchmarks. In particular, we first pre-train using the initial 50% of classes for 100 epochs. Then, we fine-tune with the remaining 50% of classes for 100 epochs, so that the fine-tuning classes and the pre-training classes do not overlap. Since the NC theory [45, 46] mainly focuses on analyzing the training loss, we only present the average training accuracy for the first 50% pre-trained classes (See Figure 2). Notably, when the fine-tuning starts, the training accuracy of pre-trained classes rapidly diminishes to zero across all datasets. As discussed in previous sections, such a catastrophic forgetting phenomenon can be directly associated with minority collapse, where the classifiers of all minority classes converge to a single vertex, when the imbalance ratio between majority and minority classes approaches infinity. Therefore, the observed decline in performance is in line with

our expectations. For completeness, we provide *the theoretical formulation of minority collapse* of fine-tuning in Appendix A and implementation details in Appendix B.

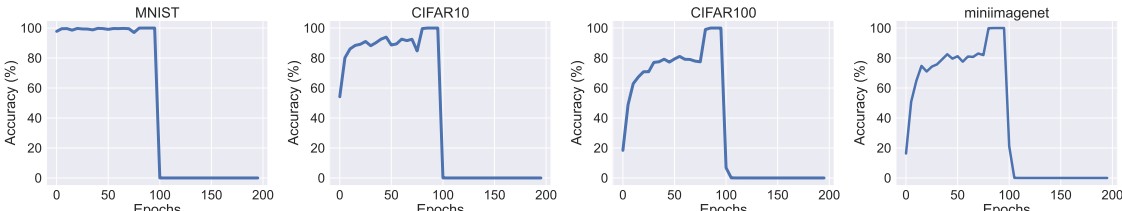

Figure 2: **Catastrophic forgetting happens in traditional classification tasks.** To corroborate the NC theory [45–48], we only plot the average *training accuracy* of the first 50% classes of MNIST, CIFAR-10, CIFAR-100, and *mini*Imagenet, respectively.

## 3.2. Fine-Tuning Contrastive Language-Image Pre-Training Network

We then fine-tune the vision encoder from the CLIP ViT-L-14 model [11], starting from a checkpoint provided by OpenAI's CLIP, available through openCLIP [12]. In our experiments, we employ the standard cross-entropy loss, consistent with the approach used in CLIP pre-training and the analysis in Neural Collapse [45, 46] as well as minority collapse [47]. Text inputs are created by concatenating labels with short descriptions. See examples in Appendix B.

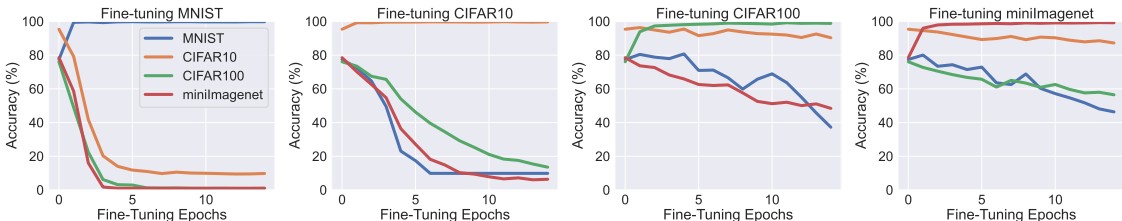

Figure 3: Accuracy of 1-14 epoch fine-tuned CLIP on MNIST, CIFAR-10, CIFAR-100, and *mini*Imagenet. Detailed accuracy numbers are presented in Table 10 of Appendix B.3.

Empirical results demonstrate that vision-language models like CLIP are susceptible to neural collapse after fine-tuning. In particular, we observe a significant rise in the in-domain performance, while the out-of-domain dataset performance begins to decline. By the time we reach 15 epochs, nearly all in-domain performance metrics have escalated to close to 99%, but the out-of-domain performance has suffered.

## 4. EMT: Evaluating Multimodal Large Language Models

Since prior MLLM evaluation frameworks [17, 25] focus on assessing cognitive reasoning [17] or hallucinations [25] rather than the catastrophic forgetting from an image classification perspective, we propose EMT, a framework for **E**valuating **M**ul**T**imodal LLM. EMT works as follows: (1) We start by inputting an image from a classification task; (2) Then we prompt the testing MLLM by asking it to classify the input images and collect its outputs via the prompt provided below, according to each dataset. (3) Next, since the output from MLLMs may not adhere to a specific format, we apply GPT-3.5 to evaluate the classification accuracy;[2] (4) Finally, we output the prediction accuracy of the testing MLLM on different datasets.

> **EMT Prompt:**
>
> What is the number/object in the image?  Please only answer a single
> number/object in [class labels].

---

[2]It is a common practice to adopt openaiAPI for evaluating the performance of different LMs, e.g., see [28, 52]. See more discussion on other potential evaluation methods in Section 7.

The detailed prompts for predictions and evaluations for each dataset are provided in Appendix C.1.

## 4.1. Catastrophic Forgetting in Open-Source MLLMs

In this subsection, we initially apply EMT to assess four MLLMs: LLaVA [7], Otter [9], Instruct-BLIP [10], and LENS [26]. As shown in Figure 4, most of the tested open-source MLLMs suffer from catastrophic forgetting by failing to retain a similar classification performance, compared to the zero-shot classification outcome of their respective vision encoders. A notable exception is `InstructBLIP-7b`, which performs slightly better on the CIFAR-10 dataset. Despite `InstructBLIP` slightly performing better than its base vision model, `InstructBLIP` cannot achieve similar performance in CIFAR-100 and *mini*Imagenet, compared to `LLaVA` and `Otter`.[3] It may seem surprising that most of the tested MLLMs fail to retain similar performance of their foundational vision models, but such a performance degradation can be anticipated in hindsight. This performance degradation may stem from the fact that classifications of MNIST, CIFAR-10, CIFAR-100, and *mini*Imagenet are not incorporated into the training dataset of the evaluated MLLMs.[4]

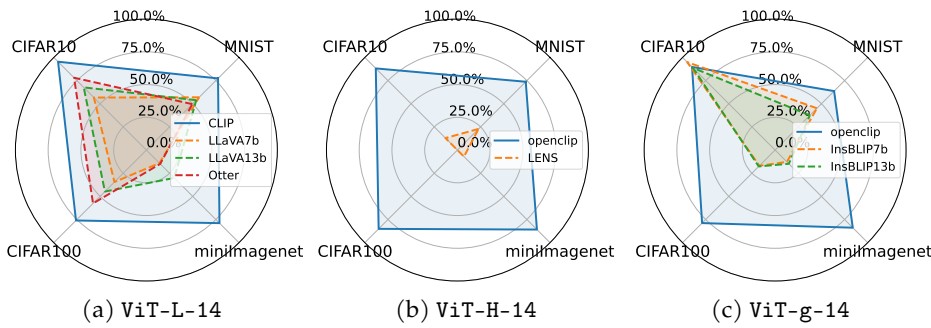

(a) `ViT-L-14`     (b) `ViT-H-14`     (c) `ViT-g-14`

Figure 4: EMT evaluation accuracy of different MLLMs on MNIST, CIFAR-10, CIFAR-100, and *mini*Imagenet, against the zero-shot performance of their vision encoders. Models are grouped according to their underlying vision encoder architecture. Detailed accuracy numbers are presented in Table 2 in Appendix C.3.

## 4.2. Analyzing Failure Modes of MLLMs

After checking the outputs of different models using our EMT prompt, we have identified three major issues causing performance degradation: incorrect prediction, intrinsic hallucination, and extrinsic hallucination. It is evident that MLLMs could produce incorrect predictions, just like classifiers. In the example shown below, `LLaVA-7B` incorrectly predicts "0" as "8" in the MNIST classification.

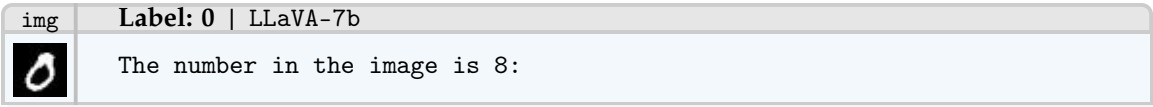

Aside from incorrect prediction, the performance is also significantly impacted by hallucination [27, 53] – the tested MLLMs sometimes generate seemingly related but incorrect or unverifiable contents. Ji et al. [27] further characterizes hallucinations into two distinct categories: *intrinsic* and *extrinsic* hallucinations. Intrinsic hallucinations are defined as instances in which the generated output directly contradicts the source content. Extrinsic hallucinations, on the other hand, are those where the output bears no verifiable connection to the original source content.

**Intrinsic Hallucination.** Our EMT prompt has identified intrinsic hallucinations within the tested MLLMs. One example can be drawn from asking LENS to perform a classification on CIFAR-10:

---

[3]We hypothesize that the performance variations amongst these MLLMs are attributable to differences in their training methodologies. However, the precise causes contributing to the performance discrepancy in these open-source MLLMs are beyond the scope of this research.

[4]For completeness, we leave the detailed discussion of different datasets adopted by each tested MLLMs in Appendix C.2. We also have some examples of the outputs by EMT prompt in Appendix C.4

| | |
|---|---|
| 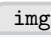 **Label: horse** \| LENS | |
| 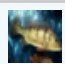 `airplane, automobile, bird, cat, deer, dog, frog, horse,` | |

It is important to note that EMT prompt explicitly instructed the testing MLLM to identify only a single object within all class labels. Despite these clear instructions, LENS still produces an intrinsically hallucinated output - `airplane, automobile, bird, cat, deer, dog, frog, horse,`, an answer that contains multiple labels.

**Extrinsic Hallucination.** In addition to intrinsic hallucination, we have also discovered extrinsic hallucinations when applying `InstructBLIP` to classify CIFAR-100:

| | |
|---|---|
| **Label: aquarium_fish** \| InstructBLIP-7b | |
| `a picture of a fish in a tank` | |

In the example provided above, while the generated output text partially includes the label "aquarium fish", it also exhibits additional descriptors that are not only challenging to verify, but also extraneous to the original request outlined by the prompt.

**Base LMs are Important.** Among all the tested MLLMs, Figure 4 shows that LENS achieves the worst performance, compared to all other models, in each individual task and overall performance. Considering that `ViT-H-14`, the underlying vision encoder of LENS, does not exhibit a significant performance shortfall, we hypothesize that the observed performance gap is attributed to the base LM. This is because Otter, LLaVA, and InstructBLIP all adopt the LLaMA model [14], while LENS uses the Flan-T5 model [13], which is less powerful than LLaMA. Nonetheless, our results do not necessarily imply that larger LMs consistently yield superior performance, as our experiments have revealed varying outcomes. For instance, although `LLaVA-13b` generally surpasses `LLaVA-7b`, `InstructBLIP-13b` does not demonstrate superiority over `InstructBLIP-7b`. Therefore, we believe that additional experiments are required to conclusively determine whether larger LMs improve the integration of vision and text data in MLLMs.

## 5. EMT on Multimodal Large Language Models Fine-Tuning

Equipped with EMT, we now investigate the hallucinations in MLLM fine-tuning. We use `LLaVA-7b` and `LLaVA-13b` as our based MLLM for fine-tuning. And we conduct fine-tuning experiments on MNIST, CIFAR-10, CIFAR-100, and *mini*Imagenet, respectively. All of our fine-tuning experiments were conducted based on the LLaVA model released on July 4th, 2023.[5]

**Linear and LoRA Fine-Tuning** As discussed by Liu et al. [7], the LLaVA model contains a frozen base vision encoder $g(\cdot)$ and a pre-trained LLM $f_\phi(\cdot)$ parameterized by $\phi$. For an input image $X_\text{v}$, LLaVA first maps $X_\text{v}$ into a visual feature vector $Z_\text{v}$ by applying the visual encoder $Z_\text{v} = g(X_\text{v})$. Then, LLaVA applies a linear adapted layer $W$, that maps the visual features into text feature spaces $H_\text{v} = W \cdot Z_\text{v}$, and concatenate $H_\text{v}$ with the embedding of language queries $H_\text{q}$ into a visual and text embedding vector $[H_\text{v}, H_\text{q}]$. Finally, LLaVA feeds $[H_\text{v}, H_\text{q}]$ as the input to the pre-trained LLM $f_\phi(\cdot)$ to obtain responses. As for specific fine-tuning methods: (1) Linear fine-tuning method *only* fine-tunes the linear adapter layer $W$; (2) LoRA fine-tuning method fine-tunes the linear adapter layer $W$ *and* the pre-trained LLM $f_\phi(\cdot)$ with LoRA [54].

### 5.1. Experimental Setup and Overview

Given that LLaVA relies on visual and language instruction data for training and fine-tuning processes, we have manually reformatted several datasets, namely MNIST, CIFAR-10, CIFAR-100, and *mini*Imagenet to comply with the required format for fine-tuning. For more detailed information on the format of the fine-tuning data used, as well as the specifics of the LLaVA fine-tuning process,

---

[5]See this git commit.

please refer to Appendix D.1. All of our fine-tuning experiments were conducted using 2 Nvidia A100 GPUs. We fine-tune LLaVA-7b and LLaVA-13b using linear and LoRA [54] fine-tuning respectively, due to the limitation of computational resources, we cannot afford to fine-tune the entire LLaMA model. We first report the EMT evaluated accuracy of fine-tuned LLaVA-7b and LLaVA-13b after 3 epochs of linear and LoRA fine-tuning in Figure 5. To assess accuracy variations during training, we then report the EMT evaluation results from 1-3 fine-tuning epochs in Figure 6 and 7.

## 5.2. Excessive Fine-Tuning Causes Forgetting

We first present the 3-epoch fine-tuning results in Figure 5. While LLaVA's performance indeed improves on the fine-tuning dataset, Figure 5 unveils a critical issue of MLLM fine-tuning:

*Fine-tuning MLLM on one dataset decreases the performance on another non-fine-tuning dataset.*

This phenomenon, though not unexpected, is noteworthy. As the model doesn't have exposure to datasets other than the one it has been fine-tuned on, it stands to reason that a similar effect to catastrophic forgetting would be observed, as discussed previously in Section 4.1.

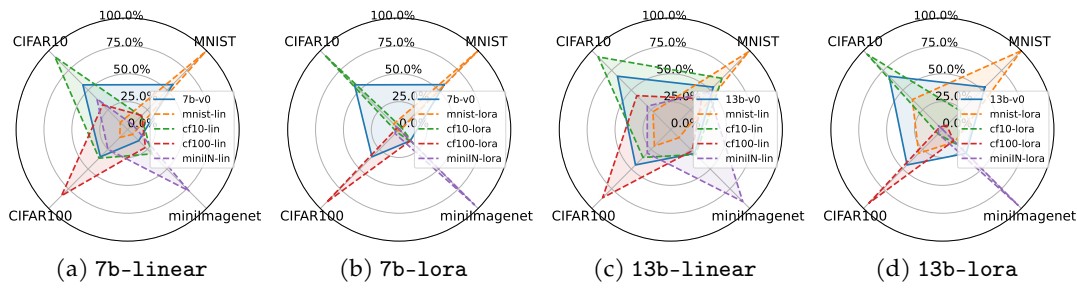

| (a) 7b-linear | (b) 7b-lora | (c) 13b-linear | (d) 13b-lora |

Figure 5: EMT evaluation accuracy of 3-epoch fine-tuned LLaVA-7b and LLaVA-13b on MNIST, CIFAR-10, CIFAR-100, and *mini*Imagenet, against the zero-shot performance of their vision encoders. Detailed accuracy numbers are presented in Table 9 of Appendix D.5.

As we examine the output from fine-tuned LLaVA, we discover that

*Fine-tuning MLLM causes hallucinations, by outputting texts that are related to its fine-tuned dataset while ignoring the question related to its original prompt.*

To further illustrate this phenomenon, we provide explicit examples of classifying the LLaVA-7b and LLaVA-13b, which have been fine-tuned on different datasets using the EMT prompt.

> **EMT Prompt:**
>
> ```
> What is the object in the image?  Please only answer a single object in
> airplane, automobile, bird, cat, deer, dog, frog, horse, ship, truck.
> ```

| img | **Label: airplane** | LLaVA-7b-lora-ft-cifar10 |
| --- | --- |
| 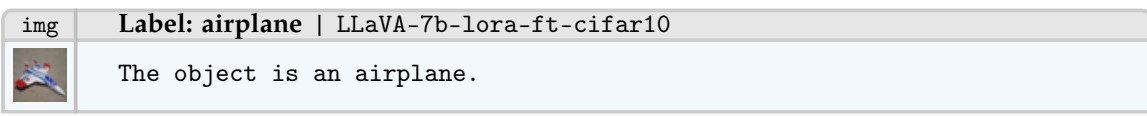 | ```
The object is an airplane.
``` |

The earlier demonstration illustrates that, when the CIFAR-10 fine-tuned model is tested on CIFAR-10, LLaVA indeed successfully identifies the object. Nevertheless, the LLaVA model begins to hallucinate in CIFAR-10 classifications after being fine-tuned on other datasets.

| img | **Label: airplane** | LLaVA-7b-lora-ft-mnist |
| --- | --- |
| | ```
The airplane is 8.
``` |

In the preceding example, the classification of CIFAR-10 through an MNIST fine-tuned model, the model not only partially generates the keyword "airplane", but concurrently produces hallucinated outputs by yielding the representation of the number "8". Similar phenomena are also observed in the CIFAR-100 and *mini*Imagenet fine-tuned models. Specifically, these fine-tuned models begin to hallucinate by predicting "airplane" as classes that bear resemblance or are related to an "airplane", such as "butterfly" and "aircraft carrier" in the CIFAR-100 and *mini*Imagenet models, respectively.

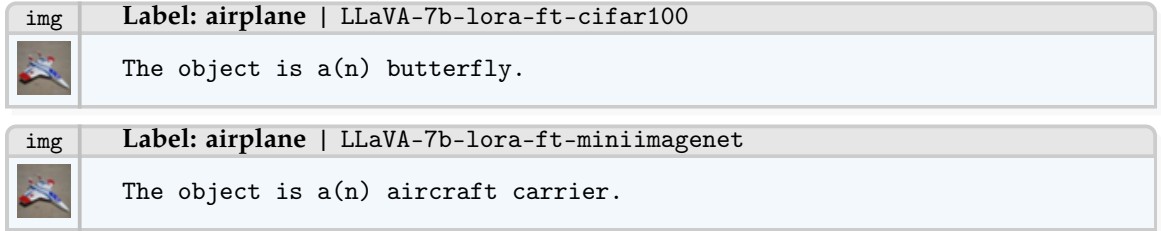

For completeness, we attach additional outputs of different fine-tuned LLaVA models in Appendix D.2 for further reference.

### 5.3. Moderate Fine-Tuning is Beneficial

In the preceding subsection, we have demonstrated that 3-epoch fine-tuned LLaVA achieves superior performance on the fine-tuned dataset, at the expense of generating hallucinated texts when tested on other datasets. However, this outcome does not necessarily imply that fine-tuning undermines the performance. Notably, we actually observe performance improvement on non-fine-tuned datasets. For instance, as shown in Figure 5, `LLaVA-7b` exhibits improved performance on *mini*Imagenet after 3 epochs of fine-tuning on CIFAR-10. To better understand the generalizability in fine-tuning, we conduct fine-tuning experiments on all four datasets for 3 epochs and report their accuracy at each epoch.

**Fine-Tuning Adapter Improves Feature Alignments.** As illustrated in Figure 6, we observe that the linear fine-tuned LLaVA achieves generalization performance upon being fine-tuned on RGB datasets, namely, CIFAR-10, CIFAR-100, and *mini*Imagenet. Given that linear fine-tuning only affects the linear projection layer connecting visual features to the text embedding space, Figure 6 implies that early-stage fine-tuning contributes to the enhancement of alignment between visual and textual features. However, in subsequent fine-tuning epochs (2-3), LLaVA starts to overfit the fine-tuning dataset by generating hallucinated texts.

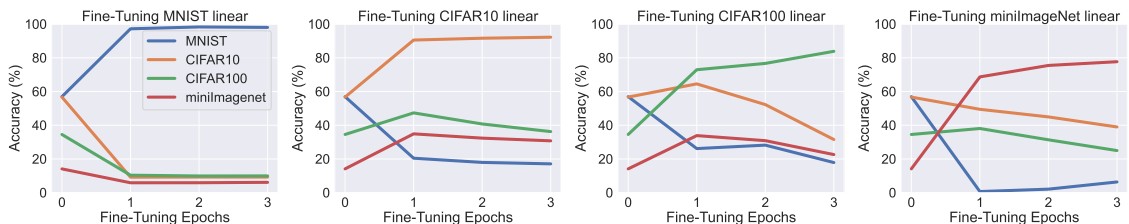

Figure 6: EMT evaluation accuracy of 1-3 epoch linear fine-tuned `LLaVA-7b` on MNIST, CIFAR-10, CIFAR-100, and *mini*Imagenet. Detailed accuracy numbers are presented in Table 10.

**Fine-Tuning LLM and Adapter Causes Hallucinations.** Contrary to the linear fine-tuning, Figure 7 implies that jointly fine-tuning both the LLM and the linear adapter directly causes overfitting on the fine-tuning dataset. This is evidenced by the significant degradation in the LoRA fine-tuned model's performance on the non-fine-tuning datasets after just a single epoch of training.

## 6. Conclusions

In this paper, we have studied how fine-tuning affects catastrophic forgetting in MLLMs. To quantitatively evaluate this issue, we propose EMT, a framework for evaluating the fine-tuning perfor-

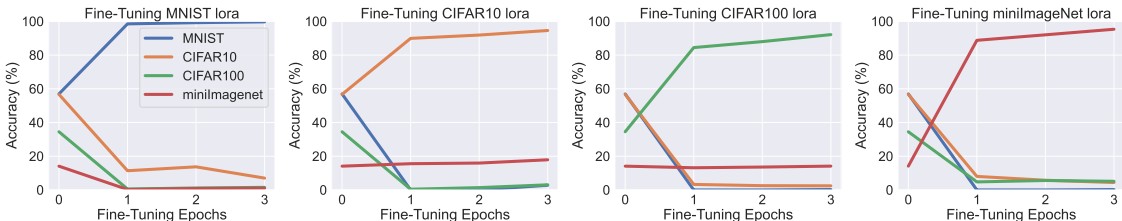

Figure 7: EMT evaluation accuracy of 1-3 epoch LoRA fine-tuned `LLaVA-7b` on MNIST, CIFAR-10, CIFAR-100, and *mini*Imagenet. Detailed accuracy numbers are presented in Table 11.

mance of MLLMs. We then conduct extensive experiments in fine-tuning LLaVA, an MLLM, and apply EMT to evaluate the performance of different fine-tuned LLaVA models. We have discovered that: (1) Almost all the open-source MLLMs tested in this paper fail to achieve a similar level of accuracy, compared to the zero-shot performance of their base vision encoder; (2) After excessive fine-tuning on one dataset, LLaVA's performance on non-fine-tuning datasets deteriorate as it starts to overfit and hallucinate; (3) Moderate fine-tuning actually improves the performance of LLaVA on similar tasks, as fine-tuning helps visual and text feature alignment in the early-stage.

# 7. Discussions and Future Work

**Dataset Diversity is Important for Fine-Tuning.** Figure 6 shows that LLaVA fine-tuned on CIFAR-10, CIFAR-100, and *mini*Imagenet for one epoch, could generalize to the other two datasets, while fine-tuning LLaVA on MNIST leads to performance degradation on all remaining datasets. This observation implies that having a diverse fine-tuning dataset is important. This is because a more diverse dataset will have features of more modes, hence making the fine-tuned MLLMs suffer less from catastrophic forgetting.

**Catastrophic Forgetting Beyond Image Classifications.** As a starting point, we only study the catastrophic forgetting in MLLM from the image classification perspective, since it is a standard classification problem. In the future, we believe similar evaluation methods can be developed for other scenarios, such as reducing bias towards unsafe outputs [39], degrading visual localization reasoning capabilities [8], or even hallucinations [25].

**Post-processing the Outputs.** Note that in step (3) of EMT, using the `openaiAPI` is not the only solution for evaluating the correctness of the outputs generated by MLLMs. In the future, there are several solutions. (1) Utilize a sentence embedding model. $N$ formatted ground truth phrases can be fed into a sentence embedding model such as CLIP text encoding resulting in $N$ ground truth embedding $\{e_i\}$, where $i \in \{1, \cdots, N\}$. Given a generated text $y$ for a test sample, we can feed its CLIP text embedding $e(y)$ and compute the matching ground truth $i$ using $\arg\min_i \|e_i - e(y)\|_2$. (2) One can also hard code (such as finding the existence of the label names) the decision criteria for dealing with hallucination. Note that finding a perfect post-processing method for EMT is not easy, as the labels from different datasets may have many synonyms. For example, when evaluating LLaVA on the label `African_hunting_dog` in *mini*Imagenet, it is hard to determine whether a prediction of "dog" should be correct or not. Hence, we believe such confusion in synonyms should also be taken into consideration in the future when building post-processing methods.

# 8. Acknowledgement

We want to thank Sergey Levine from UC Berkeley and Carl Vondrick from Columbia University, for the early discussion during the preparation of this paper. We would also like to thank Haotian Liu from the University of Wisconsin-Madison for suggestions in setting up the LLaVA experiments. We would also like to thank Samuel Ainsworth and Yuning Chai from the Cruise AI research team for their insightful discussion and suggestions. YZ and YM acknowledge support from the joint Simons Foundation-NSF DMS grant #2031899, the ONR grant N00014-22-1-2102, and the Tsinghua-Berkeley Shenzhen Institute (TBSI) Research Fund. Yi Ma also acknowledges support from the University of Hong Kong.

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

# Appendices

## A. Neural Collapse

**Notations.** We use bold capital letters (e.g., $\boldsymbol{M}$) to denote matrices and bold smaller case letters (e.g., $\boldsymbol{v}$) to denote vectors. And for a vector $\boldsymbol{v} \in \mathbb{R}^d$, we use $v_i, \forall i \in [d]$ to denote its $i^{\text{th}}$ entry. We consider a $K$ class classification setting, where $\forall k \in [K]$, we use $\boldsymbol{y}_k$ to denote the $k^{\text{th}}$ one-hot vector: $\boldsymbol{y}_k = [\underbrace{0, \ldots, 0}_{k-1 \ 0\text{s.}}, 1, 0, \ldots, 0]^\top$. We then use $\boldsymbol{W} = [\boldsymbol{w}_1, \boldsymbol{w}_2, \ldots, \boldsymbol{w}_K]^\top \in \mathbb{R}^{K \times d}$ to denote the weight matrix of the *last fully connected layer of a neural network* and $\boldsymbol{w}_k$ is the $k^{\text{th}}$ row vector of $\boldsymbol{W}$, $\forall k \in [K]$. Next, we use $\boldsymbol{H} = [\boldsymbol{h}_{k,i} : 1 \le k \le K, i \le n_k] \in \mathbb{R}^{d \times N}$ to denote the matrix of all feature vectors corresponding to all inputs, where $n_k$ represents the number of samples of the $h^{\text{th}}$ class and $N$ is the total number of samples. Our analysis will focus on the cross-entropy loss w.r.t. the one hot label vectors $\boldsymbol{y}_k, \forall k \in K, \forall \boldsymbol{z} \in \mathbb{R}^d$: $\mathcal{L}(\boldsymbol{z}, \boldsymbol{y}_k) = -\log \left[ \frac{\exp(z_k)}{\sum_{k'=1}^{K} \exp(z'_k)} \right]$.

### A.1. Preliminary Results of Neural Collapse and Minority Collapse

**Neural Collapse.** The *neural collapse* (NC) phenomenon is first observed in [45] when minimizing the cross-entropy loss for classification. NC characterizes the optimal geometric structures of model classifiers and features in balanced multi-class classification, leveraging such structures during training has been proven effective in mitigating the challenges of imbalanced data training [55, 56], as well as incremental learning settings [23]. Theoretical insights have also emerged from various studies [47–50] to explore the understanding of data imbalance training from the NC perspective. Fang et al. [47] introduced Minority Collapse.

**Minority Collapse.** Fang et al. [47] extends the theoretical results of NC into the imbalance training setting, revealing that when the *imbalance ratio* of the dataset improves, the feature vectors of *the minority class will converge to one single vector*, which is known as *minority collapse*. Mathematically, Fang et al. [47] assumes the majority classes and minority classes have $n_a, n_b$ samples per class, respectively. Under such imbalance data model, Fang et al. [47] shows that when the imbalance ratio $n_a/n_b \to \infty$, any pair of the weight vectors $\boldsymbol{w}_k^\star, \boldsymbol{w}_{k'}^\star$ from the minority classes will satisfy $\lim_{n_a/n_b \to \infty} \boldsymbol{w}_k^\star - \boldsymbol{w}_{k'}^\star = \boldsymbol{0}$ at convergence during cross-entropy training. We leave the formulation and main results of Fang et al. [47] in Appendix A.3 for completeness.

**Data Imbalance and the SELI geometry.** While Fang et al. [47] aims to characterize the geometric attributes of feature representation and classifier weights in an asymptotic manner, particularly as the imbalance ratio tends towards infinity, the work by Thrampoulidis et al. [48] offers a more comprehensive exploration of the geometry within the final layer. The latter paper introduces a novel geometric construct known as the Simplex Encoded-Labels Interpolation (SELI) geometry. This geometry characterizes the optimal logit arrangement within the constraints of the unconstrained feature model assumption [57], irrespective of data imbalance. Moreover, the study demonstrates that this framework transitions to the simplex Equiangular Tight Frame (ETF) structure in scenarios of balanced data and aligns with the phenomenon of minority collapse under conditions of asymptotic imbalance.

**Theoretical Formulation of NC.** NC illustrates that the last-layer features and classifiers assuming *each class has the same training samples*, namely $n_k = n_{k'}, \forall k, k' \in [K]$. As also discussed in later works [46, 47], one particular NC phenomenon reveals that the last-layer classifiers (weight matrix $W$) will be maximally contrastive [58]. Mathematically, this implies that the cosine similarity $\frac{\langle w_i, w_j \rangle}{\|w_i\| \|w_j\|}$ between any pair of the row vectors $w_i, w_j, \forall i, j \in [K], i \neq j$ of $W$ *reaches the largest value* for $K$ equiangular vectors from solving the regularized cross-entropy minimization problem: reformulates the $\ell^2$-regularized cross-entropy minimization problem:

$$\min_{W_{\text{full}}} \frac{1}{N} \sum_{k=1}^{K} \sum_{i=1}^{n_k} \mathcal{L} \left( f \left( x_{k,i}; W_{\text{full}} \right), y_k \right), \tag{A.1}$$

where $\{x_{k,i}\}_{i=1}^{n_k}$ denotes the training examples in the $k^{\text{th}}$ class, and

$$f(x; W_{\text{full}}) = b_L + (W_L \sigma (W_{L-1} \sigma (\cdots \sigma (b_1 + W_1 x) \cdots)))$$

is the output of an $L$ layer fully connected neural network, and $W_{\text{full}} = \{W_1, W_2, \ldots, W_L\}$ denotes the weights of all $L$ layers.

## A.2. A Minority Collapse Perspective of Fine-Tuning

We consider the following setting for the pre-training and fine-tuning using the cross-entropy loss. In the pre-training phase, we only assume access to a certain classes of labels $\mathcal{U}_{\text{pt}}, \mathcal{U}_{\text{ft}} \subsetneq [K]$ during the pre-training and fine-tuning phase respectively. Since we assume both $\mathcal{U}_{\text{pt}}$ and $\mathcal{U}_{\text{ft}}$ are strict subset of $[K]$, hence in both the pre-training and fine-tuning phases, the classification problems naturally become imbalanced problems, with the imbalance ratio of $\infty$. This is because the missing classes $\bar{\mathcal{U}}_{\text{pt}} := [K] \backslash \mathcal{U}_{\text{pt}}, \bar{\mathcal{U}}_{\text{ft}} := [K] \backslash \mathcal{U}_{\text{ft}}$ during the pre-training and fine-tuning phases have 0 samples. Applying the main result from minority collapse [47], we know that the classification accuracy of the classes that *appear in the pre-training phase but are absent during fine-tuning* ($\forall k \in \mathcal{U}_{\text{pt}} \cap \bar{\mathcal{U}}_{\text{ft}}$), will degrade after fine-tuning since the weight vectors for these classes will collapse to the same vector ($w_k^\star - w_{k'}^\star = 0$). In the next section, we will demonstrate such degradation in pre-training and fine-tuning in image classification, fine-tuning pre-trained contrastive language-image networks, and multimodal visual large language models.

## A.3. Minority Collapse

Instead of directly analyzing the cross-entropy minimization objective Eq. (A.1), the theoretical literature studies the $\ell^2$-norm regularized version since weight norm regularization methods (such as weight decay) are commonly adopted in practical deep learning training [51]:

$$\min_{W_{\text{full}}} \frac{1}{N} \sum_{k=1}^{K} \sum_{i=1}^{n_k} \mathcal{L} \left( f \left( x_{k,i}; W_{\text{full}} \right), y_k \right) + \frac{\lambda}{2} \|W_{\text{full}}\|^2. \tag{A.2}$$

Fang et al. [47] reformulates the $\ell^2$-regularized cross-entropy minimization problem Eq. A.2 into the Layer-Peeled Model by only consider the weight matrix $W$ and feature vectors $h_{k,i}$ of the last layer with certain norm constraints:

$$\min_{W, H} \frac{1}{N} \sum_{k=1}^{K} \sum_{i=1}^{n_k} \mathcal{L} \left( W h_{k,i}, y_k \right), \quad \text{s.t.} \quad \frac{1}{K} \sum_{k=1}^{K} \|w_k\|_2^2 \leq E_W, \quad \frac{1}{K} \sum_{k=1}^{K} \frac{1}{n_k} \sum_{i=1}^{n_k} \|h_{k,i}\|_2^2 \leq E_H. \tag{A.3}$$

Under the Layer-Peeled Model Eq. (A.3), Fang et al. [47] further assumes that, among the entire $K$ classes, the first $K_A$ classes are the majority classes, with $n_a$ samples per class. While the remaining $[K] \backslash [K_A]$ classes are the minority classes, with $n_b$ samples per class. We state the original results on the minority collapse as follows:

**Theorem A.1 (Thm.5 of [47])** *Assume $p \geq K$ and $n_A / n_B \rightarrow \infty$, and fix $K_A$ and $K_B$. Let $(H^\star, W^\star)$ be any global minimizer of the Layer-Peeled Model Eq. (A.3) with the cross-entropy loss. When $n_a / n_b \rightarrow \infty$, we have*

$$\lim w_k^\star - w_{k'}^\star = \mathbf{0}_p, \quad \forall K_A < k < k' \leq K. \tag{A.4}$$

### A.4. A Neural Collapse Perspective of Next Token Prediction

Note that the MLLM training [7, 9, 10] also adopts the cross-entropy loss and the outputs are performing sequential token generations, one could still treat the per-token prediction process as a sequential classification and apply the NC model to understand its behavior. Specifically, when treating the next token generation task as a prediction problem, one can view the preceding text as input for classification, and the next token as the prediction output chosen from the entire vocabulary. Similar to the previous subsections, as long as the pre-training dataset and fine-tuning dataset have different supports, one should expect to see catastrophic forgetting in multimodal model fine-tuning, similar to the aforementioned cases in image classifications.

## B. Additional Results of Fine-Tuning on Image Classification

### B.1. Experimental Details of Training ResNet

For each dataset in {MNIST, CIFAR10, CIFAR100, and *mini*Imagenet}, we initiate preprocessing to normalize each dataset by its mean and variance channel-wise and do data augmentation of RandomCrop and RandomHorizontalFlip. Then we train ResNet18 [51] for 200 epochs where we pre-train the initial 50% of classes of all datasets for 100 epochs and subsequently fine-tune the remaining 50% of classes for 100 epochs. Throughout the experiments, we use SGD optimizers with a learning rate of 0.1, momentum of 0.9, and weight decay of 5e-4. We use a learning rate step decay scheduler where we decay the learning rate by the factor of 10 every 80 epochs and we choose the batch size to be 128 for all datasets. We note that for the *mini*Imagenet dataset since we are not doing few-shot learning, we split the total 60k images into the training set (50k images) and validation set (10k images) such that both the training and validation set include the full 100 classes. During pre-training, we set the weight of the first 50% of pre-training classes to be 1, and the remaining classes to 0. Whereas during fine-tuning we set the weight of the last 50% of fine-tuning classes to 1, and the remaining classes to 0.

### B.2. Additional Results of Training ResNet

To study the catastrophic problem more comprehensively, besides the results we demonstrated in Section 3, we conducted additional experiments aimed at exploring the strategy to use a new classifier (instead of changing the criterion weights) during fine-tuning. The results of these experiments are reported in Figure 8. Notably, our findings indicate that this strategy does indeed have an impact, mitigating catastrophic forgetting to a certain extent. Moreover, we observed a correlation between the degree of catastrophic forgetting and task complexity. Specifically, in the case of MNIST, forgetting occurs but is less severe, whereas, for CIFAR100 and miniImageNet, the degree of forgetting remains similar to the original results.

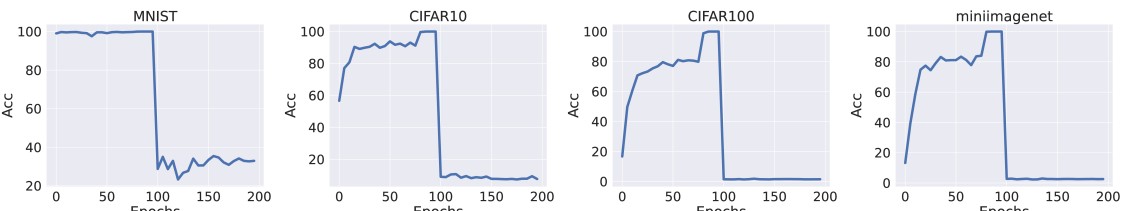

Figure 8: **Re-initialize classifier during fine-tuning helps to mitigate catastrophic forgetting slightly.** We follow the experimental setup as in Figure 2 but adopt a new classifier during fine-tuning.

Moreover, we also conducted additional experiments involving resetting the learning rate during fine-tuning as an ablation study. To be more specific, we restarted both the optimizer and the learning rate scheduler at the moment of the transition to the fine-tuning phase, and we present the results in Figure 9. As demonstrated, the training accuracy for the pre-trained classes exhibits a curve nearly identical to that depicted in the manuscript, that catastrophic forgetting still happens.

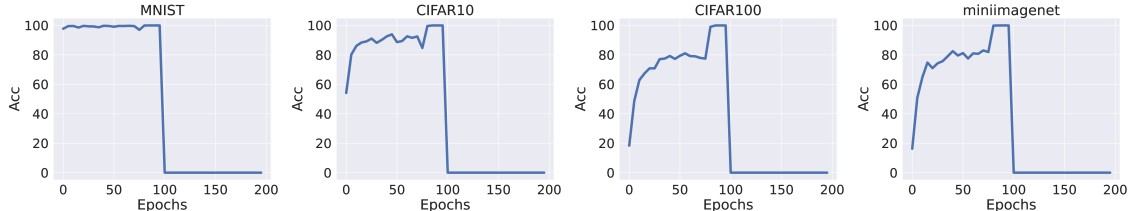

Figure 9: **Re-initialize optimizer during fine-tuning has no effect on catastrophic forgetting.** We follow the experimental setup as in Figure 2 but restart the optimizer and learning rate scheduler during fine-tuning

## B.3. Experimental Details of Fine-Tuning CLIP

**Prompts Used in Fine-Tuning and Zero-Shot Classification.** Our prompts for fine-tuning and zero-shot classification differ across datasets for better performance. For each dataset in {MNIST, CIFAR10, CIFAR100, *mini*Imagenet}, we use the following prompts:

- MNIST:

  > **Prompt:**
  >
  > ```
  > a photo of the number:  "{digit}".
  > ```

- CIFAR-10:

  > **Prompt:**
  >
  > ```
  > a photo of a {object}.
  > ```

- CIFAR-100:

  > **Prompt:**
  >
  > ```
  > a photo of a {object}.
  > ```

- *mini*Imagenet:

  > **Prompt:**
  >
  > ```
  > a photo of a {object}.
  > ```

**Training Details of Fine-Tuning CLIP.** When fine-tuning a given dataset, we use all the datasets available to fine-tune the CLIP model. We use a batch size of 128, Adam Optimizer [59]. We set the learning rate to 1e-5 and weight decay to 0.001.

## C. Evaluating Existing Models with EMT

### C.1. EMT Prompts

**Classification Prompts.** Our prompts for classification differs across datasets. For each dataset in {MNIST, CIFAR10, CIFAR100, *mini*Imagenet}, we use the following prompts:

- MNIST:

  > **EMT Prompt:**
  >
  > ```
  > What is the number in the image?  Please only answer a single
  > number in 0, 1, 2, 3, 4, 5, 6, 7, 8, 9.
  > ```

|  | Datasets | | | |
| Checkpoints | MNIST | CIFAR-10 | CIFAR-100 | *mini*Imagenet |
|---|---|---|---|---|
| openai | 77.41% | 95.36% | 76.05% | 78.49% |
| mnist-ft-5ep | 99.84% | 11.80% | 2.93% | 1.00% |
| cifar10-ft-5ep | 17.40% | 99.65% | 46.21% | 27.13% |
| cifar100-ft-5ep | 70.92% | 91.55% | 98.33% | 62.58% |
| miniimagenet-ft-5ep | 72.90% | 89.24% | 65.70% | 98.59% |
| mnist-ft-15ep | 99.76% | 9.18% | 0.99% | 1.00% |
| cifar10-ft-15ep | 9.87% | 99.41% | 11.04% | 5.67% |
| cifar100-ft-15ep | 39.64% | 85.85% | 99.27% | 46.64% |
| miniimagenet-ft-15ep | 30.93% | 87.18% | 56.39% | 98.95% |

Table 1: Zero-short performance of fine-tuned CLIP [11] on the vision model `ViT-L-14 openai` [12].

- CIFAR-10:

> **EMT Prompt:**
>
> ```
> What is the object in the image?  Please only answer a single
> object in airplane, automobile, bird, cat, deer, dog, frog, horse,
> ship, truck.
> ```

- CIFAR-100:

> **EMT Prompt:**
>
> ```
> What is the object in the image?  Please only answer a single
> object in apple, [other CIFAR-100 labels in text], worm.
> ```

- *mini*Imagenet:

> **EMT Prompt:**
>
> ```
> What is the object in the image?  Please only answer a single
> object in African hunting dog, [other miniImagenet labels in
> text], yawl.
> ```

**Evaluation Prompts.** For evaluating the prediction accuracy of the testing MLLM, we ask `gpt-3.5-turbo` with the following prompt:

> **EMT Prompt:**
>
> ```
> Please only answer the question in yes or no.  Is the "Prediction"
> correctly predicting the right "Label"?  Label:  {label}; Prediction:
> {outputs}.
> ```

For the parameters in `gpt-3.5-turbo`, we set `temperature=0.2`, `max_tokens=64`, `top_p=1`, `frequency_penalty=0`, `presence_penalty=0`.

## C.2. Fine-Tuning Dataset of Tested MLLMs

We list all datasets used by all tested MLLMs here:

- `LLaVA` is pre-trained on CC3M [60], a dataset with text-image pairs, and fine-tuned on GPT collected language image instruction-following data and ScienceQA [29], see more detailed in Section 4.2 of Liu et al. [7].

- `Otter` [9] proposes Multi-Modal In-Context Instruction Tuning (MIMIC-IT) dataset, a dataset that augments OpenFlamingo into an instruction-following format [39]. Then `Otter` uses the proposed MIMIC-IT dataset for fine-tuning. See more details in Section 3 of Li et al. [9].

- `InstructBLIP` [10] transforms different public vision-language datasets into instruction tuning format, and used the transformed instruction tuning datasets for training. See details in Section 2.1 of Dai et al. [10].

- `LENS` [26] does not explicitly use other vision-language datasets for fine-tuning, as it only proposes a framework to enhance the visual reasoning capability of a frozen LLM, by utilizing different visual modules. See details in Section 3.2 of Berrios et al. [26].

## C.3. Classification Accuracy

We present the classification accuracy of different vision models and different MLLMs in Table 2.

| Vision Model | Checkpoints | Datasets | | | |
|---|---|---|---|---|---|
| | | MNIST | CIFAR-10 | CIFAR-100 | *mini*Imagenet |
| `ViT-L-14` | CLIP | 77.36% | 95.39% | 76.04% | 78.92% |
| | LLaVA-7b | 56.96% | 56.71% | 34.53% | 14.12% |
| | LLaVA-13b | 53.84% | 67.67% | 44.79% | 30.11% |
| | Otter | 49.59% | 78.11% | 57.63% | 14.94% |
| `ViT-H-14` | openclip | 73.75% | 88.17% | 85.09% | 85.84% |
| | LENS | 22.84% | 12.97% | 0.03% | 7.08% |
| `ViT-g-14` | openclip | 63.67% | 89.72% | 78.86% | 83.84% |
| | InsBLIP-7b | 45.12% | 94.95% | 17.00% | 12.76% |
| | InsBLIP-13b | 37.49% | 88.72% | 17.70% | 14.82% |

Table 2: Prediction accuracy of `CLIP` [11], `Otter` [9], `LLaVA` [7], `openclip s34b_b88k` [12, 61], `LENS` [26], `InstructBLIP` [10], and `openclip s32b_b79k` [12, 61]. For vision encoders, `LLaVA` and `Otter` adopt `ViT-L-14` as their vision encoder; `LENS` adopts `ViT-H-14`; and `InstructBLIP` uses `ViT-g-14`. `LLaVA`, `Otter`, `LENS`, and `InstructBLIP` are generated via EMT. Total accuracy difference is calculated by summing up the performance gap between a multimodal LLM and its base vision model, for all 4 tasks.

## C.4. Hallucinated Outputs

We provide some outputs of different models by prompting with EMT in this subsection.

### C.4.1. Classifying MNIST with Otter

**EMT Prompt:**

```
What is the number in the image?  Please only answer a single number in
0, 1, 2, 3, 4, 5, 6, 7, 8, 9.
```

**Label: 2 | Otter:**

```
The object is a letter.
```

**Label: 5 | Otter:**

```
5.
```

> **Label: 7** | `Otter:`
>
> 7.

> **Label: 8** | `Otter:`
>
> 2.

> **Label: 4** | `Otter:`
>
> Number 4.

### C.4.2. Classifying CIFAR10 with LLaVA-7b

> **EMT Prompt:**
>
> ```
> What is the object in the image?  Please only answer a single object in
> airplane, automobile, bird, cat, deer, dog, frog, horse, ship, truck.
> ```

> **Label: airplane** | `LLaVA-7b:`
>
> The object in the image is a frog.

> **Label: bird** | `LLaVA-7b:`
>
> The object in the image is a large black bird.

> **Label: deer** | `LLaVA-7b:`
>
> The object in the image is a deer.

> **Label: horse** | `LLaVA-7b:`
>
> The object in the image is a horse.

> **Label: truck** | `LLaVA-7b:`
>
> The object in the image is a large airplane.

### C.4.3. Classifying CIFAR100 with LENS

> **EMT Prompt:**
>
> ```
> What is the object in the image?  Please only answer a single object in
> apple, [other CIFAR-100 labels in text], worm.
> ```

> **Label: orchid** | `LENS:`
>
> bird

> **Label: dolphin** | `LENS:`
>
> bird

> **Label: mushroom** | `LENS:`
>
> bird

> **Label: girl** | LENS:
>
> ```
> bird
> ```

> **Label: boy** | LENS:
>
> ```
> bird
> ```

### C.4.4. Classifying *mini*Imagenet with InstructBLIP

> **EMT Prompt:**
>
> ```
> What is the object in the image?  Please only answer a single object in
> African hunting dog, [other miniImagenet labels in text], yawl.
> ```

> **Label: street_sign** | InstructBLIP-7b:
>
> ```
> a street sign for st.  catherine's street
> ```

> **Label: street_sign** | InstructBLIP-7b:
>
> ```
> a blue street sign says arkansas on it
> ```

> **Label: coral_reef** | InstructBLIP-7b:
>
> ```
> a picture of a coral reef with seaweed in the foreground
> ```

> **Label: harvestman** | InstructBLIP-7b:
>
> ```
> a close up of a tree branch with a ladybug on it
> ```

> **Label: African_hunting_dog** | InstructBLIP-7b:
>
> ```
> a dog is standing in front of a forest with a sign that says walker hound
> ```

# D. Experimental Details of Fine-Tuning LLaVA

## D.1. Constructing Datasets for LLaVA Fine-Tuning and Data Usage

**Data Construction.** We construct vision and language instruction data in the following format for LLaVA fine-tuning.

```
"image":  "airplane/2604.jpg"
"conversations":
"from":  "human", "value":  "What is the object in the image?  <image>"
"from":  "gpt", "value":  "The object is a(n) airplane."
```

Note that the above example only illustrate one data point from the class airplane of CIFAR-10, we also similarly construct other data for MNIST, CIFAR-100, and *mini*Imagenet in our experiments, respectively.

**Data Usage.** During fine-tuning, we only fine-tune LLaVA model with training data in all dataset. EMT evaluations are only conducted using 20% random selected training data.

## D.2. Additional Hallucinated Outputs of Fine-Tuning LLaVA

In this section, we provide additional examples to compare the outputs of LLaVA with linear and Lora fine-tune. In particular, we want to illustrate that linear fine-tuning LLaVA on RGB images dataset will still preserve some of LLaVA's capability in binding visual and language data, while lora fine-tuning LLaVA directly lead to overfitting.

### D.2.1. Classifying CIFAR10 with 1-epoch fine-tuned LLaVA-7b linear on miniImagenet

**EMT Prompt:**

```
What is the object in the image?  Please only answer a single object in
airplane, automobile, bird, cat, deer, dog, frog, horse, ship, truck.
```

**Label: airplane** | LLaVA-7b-linear-ft-miniimagenet:

```
The object is a(n) airplane.
```

**Label: cat** | LLaVA-7b-linear-ft-miniimagenet:

```
The object is a(n) cat.
```

**Label: deer** | LLaVA-7b-linear-ft-miniimagenet:

```
The object is a(n) deer.
```

**Label: frog** | LLaVA-7b-linear-ft-miniimagenet:

```
The object is a(n) frog.
```

**Label: truck** | LLaVA-7b-linear-ft-miniimagenet:

```
The object is a(n) automobile.
```

### D.2.2. Classifying CIFAR10 with 1-epoch fine-tuned LLaVA-7b lora on miniImagenet

**EMT Prompt:**

```
What is the object in the image?  Please only answer a single object in
airplane, automobile, bird, cat, deer, dog, frog, horse, ship, truck.
```

**Label: airplane** | LLaVA-7b-lora-ft-miniimagenet:

```
The object is a(n) aircraft carrier.The object is a(n) aircraft carrier.
```

**Label: cat** | LLaVA-7b-lora-ft-miniimagenet:

```
The object is a(n) black-footed ferret.
```

**Label: deer** | LLaVA-7b-lora-ft-miniimagenet:

```
The object is a(n) white wolf.
```

**Label: frog** | LLaVA-7b-lora-ft-miniimagenet:

```
The object is a(n) rhinoceros beetle.
```

```
Label: truck | LLaVA-7b-lora-ft-miniimagenet:

    The object is a(n) garbage truck.
```

## D.3. More Analysis on Hallucination after Fine-tuning

We further conduct some experiments to analyze the "hallucinations" during fine-tuning. In particular, we found that the visual information is preserved after fine-tuning. More specifically, we observed that the ground truth label in the testing dataset will be mostly predicted to a few labels of the fine-tuning datasets, which are "visually similar" to the ground truth class.

We provide several examples in the following. The testing dataset is *mini*Imagenet, and the testing model is fine-tuned on CIFAR-10, using LoRa, for 3 epochs. We provide the top 3 predicted labels as well as the percentage of the appearance.

| Ground Truth Label | Top 3 Predictions |
| --- | --- |
| African Hunting Dog | dog: 62.12%, deer: 24.24%, bird: 6.06% |
| Arctic Fox | dog: 73.53%, cat: 17.65%, deer: 4.41% |
| French Bulldog | dog: 94.64%, cat: 3.57%, deer: 0.89% |
| Gordon Setter | dog: 94.81%, bird: 1.48%, horse: 1.48% |
| Ibizan Hound | dog: 88.97%, horse: 8.82%, deer: 1.47% |
| Newfoundland | dog: 93.69%, horse: 2.70%, ship: 0.90% |
| Saluki | dog: 85.19%, horse: 11.11%, bird: 2.78% |
| Tibetan Mastiff | dog: 98.28%, horse: 1.72% |
| Walker Hound | dog: 96.67%, horse: 1.67%, deer: 0.83% |
| Aircraft Carrier | ship: 75.21%, The object is an airplane: 24.79% |

Table 3: Top 3 Predictions for Ground Truth Labels after fine-tuning on CIFAR-10.

From the examples shown above, we can see that for all selected ground truth labels in *mini*Imagenet, the fine-tuned model hallucinates by producing labels in CIFAR-10 that a "visually similar" to each ground truth. E.g., African_hunting_dog → Dog, Deer; Arctic_fox → Dog, Cat; French_bulldog → Dog, Cat.

Note that we only provide the prediction percentages of 10 classes in one setting (testing miniimagenet on a CIFAR10 fine-tuned model), due to the space limitation. But the hallucinated outputs follow this pattern: fine-tuned LLaVA will generate the labels in the fine-tuned dataset, which are most "visually similar" to the ground truth label being tested.

## D.4. Additional Study on post-processing

As we have discussed in Section 7, we further conducted experiments that use the similarity between the CLIP text features for the outputs and labels. In particular:

1. We applied the OpenAI CLIP text embedding [11] to extract the text feature vector $e$ of the output "The object is a(n) lion." into a text embedding feature.

2. Then we also followed OpenAI CLIP text embedding to tokenize the labels of CIFAR10 into $e(1), e(2), \ldots .e(10)$.

3. We outputed the label $i \in [10]$, whose feature embedings has the smallest $\ell^2$ distance with the text embedding feature of $e$.

We report the results in Table 4, Table 5, Table 6, Table 7, Table 8.

|            | Datasets |          |           |              |
|-----------:|:--------:|:--------:|:---------:|:------------:|
| Checkpoints | MNIST   | CIFAR-10 | CIFAR-100 | *mini*Imagenet |
| llava-7b-v0  | 46.13%  | 68.10%   | 44.87%    | 28.35%       |
| llava-13b-v0 | 40.69%  | 78.62%   | 46.40%    | 34.27%       |

Table 4: Zero-shot Classification Performance, post-processing with CLIP features

|             | Datasets |          |           |              |
|------------:|:--------:|:--------:|:---------:|:------------:|
| Checkpoints | MNIST    | CIFAR-10 | CIFAR-100 | *mini*Imagenet |
| mnist-linear  | 98.89% | 78.94% | 58.21% | 50.83% |
| c10-linear    | 66.34% | 92.16% | 33.08% | 21.99% |
| c100-linear   | 17.20% | 64.60% | 86.39% | 27.52% |
| miniIN-linear | 29.70% | 64.88% | 38.05% | 90.40% |
| mnist-lora    | 99.85% | 77.29% | 58.36% | 43.22% |
| c10-lora      | 22.18% | 95.49% | 9.13%  | 8.58%  |
| c100-lora     | 10.49% | 41.04% | 93.17% | 17.23% |
| miniIN-lora   | 9.55%  | 44.05% | 18.92% | 95.77% |

Table 5: 3-Epoch Fine-Tuned Classification Performance of llava-13b-v0, post-processing with CLIP features

With the new clip text feature post-processing method, we observed a similar phenomena as the original submission. In particular: Fine-tuning on one dataset causes catastrophic forgetting on other datasets LoRa fine-tuning causes more severe catastrophic forgetting than linear fine-tuning. Overall, using the CLIP text feature, we observe the similar catastrophic forgetting phenomenon. Note that some of accurcracies using the new method are higher than the post-processing method using OpenAI API, because the CLIP text feature post processing will still make a correct prediction, even when the output sentence is logically incorrect. For the example provided in Section 5.2: label "airplane", when the output is "The airplane is 8.", ChatGPT will classify result as "No", since the output "The airplane is 8." is not making a classification.

But this result does not indicate we shall in general prefer CLIP embedding to ChatGPT or vice versa, since CLIP embedding only works for selecting the "most similar labels", while sometimes ignoring the correctness of the output. On one hand, the clip embedding is designed for classification will still classify "The airplane is 8." to "airplane". On the other hand clip embeddings is perhaps a reasonable option for classification task and economically friendly (as it does not query the openai API).

## D.5. Classification Accuracy of Fine-Tuning LLaVA

| Checkpoints | Datasets | | | |
|---|---|---|---|---|
| | MNIST | CIFAR-10 | CIFAR-100 | *mini*Imagenet |
| 1ep-linear | 97.22% | 80.60% | 58.87% | 49.85% |
| 2ep-linear | 98.30% | 81.09% | 59.93% | 49.11% |
| 3ep-linear | 98.03% | 80.87% | 59.04% | 49.29% |
| 1ep-lora | 98.52% | 39.84% | 22.46% | 16.18% |
| 2ep-lora | 99.24% | 55.79% | 35.75% | 24.78% |
| 3ep-lora | 99.71% | 61.95% | 42.42% | 29.03% |

Table 6: 1-3 Epoch Fine-Tuned Classification Performance of llava-7b-v0 on Mnist, post-processing with CLIP features

| Checkpoints | Datasets | | | |
|---|---|---|---|---|
| | MNIST | CIFAR-10 | CIFAR-100 | *mini*Imagenet |
| 1ep-linear | 21.47% | 90.72% | 45.43% | 27.08% |
| 2ep-linear | 18.77% | 91.69% | 39.53% | 26.15% |
| 3ep-linear | 17.91% | 92.25% | 34.77% | 24.77% |
| 1ep-lora | 9.93% | 89.90% | 3.09% | 3.45% |
| 2ep-lora | 9.64% | 91.90% | 3.96% | 3.57% |
| 3ep-lora | 12.78% | 94.59% | 6.06% | 6.79% |

Table 7: Classification Performance of llava-7b-v0 on CIFAR-10, post-processing with CLIP features

| Checkpoints | Datasets | | | |
|---|---|---|---|---|
| | MNIST | CIFAR-10 | CIFAR-100 | *mini*Imagenet |
| 1ep-linear | 19.32% | 71.10% | 71.56% | 28.93% |
| 2ep-linear | 22.17% | 58.40% | 76.05% | 24.63% |
| 3ep-linear | 24.20% | 45.30% | 80.73% | 19.50% |
| 1ep-lora | 9.80% | 43.17% | 84.45% | 18.06% |
| 2ep-lora | 9.63% | 43.92% | 88.01% | 17.91% |
| 3ep-lora | 10.07% | 41.06% | 92.18% | 18.24% |

Table 8: Classification Performance of llava-7b-v0 on CIFAR-100, post-processing with CLIP features

| Checkpoints | Datasets | | | |
|---|---|---|---|---|
| | MNIST | CIFAR-10 | CIFAR-100 | *mini*Imagenet |
| 7b-v0 | 56.96% | 56.71% | 34.53% | 14.12% |
| 7b-linear-ft-mnist | 98.03% | 9.26% | 9.96% | 6.14% |
| 7b-linear-ft-cifar10 | 17.10% | 92.23% | 36.24% | 30.76% |
| 7b-linear-ft-cifar100 | 17.88% | 31.54% | 83.86% | 22.61% |
| 7b-linear-ft-miniImagenet | 6.36% | 38.99% | 24.99% | 77.69% |
| 7b-lora-ft-mnist | 99.71% | 7.03% | 1.55% | 1.04% |
| 7b-lora-ft-cifar10 | 2.80% | 94.59% | 3.06% | 17.90% |
| 7b-lora-ft-cifar100 | 0.26% | 2.47% | 92.15% | 14.14 % |
| 7b-lora-ft-miniImagenet | 0.24% | 4.41% | 5.14% | 95.34% |
| 13b-v0 | 53.84% | 67.67% | 44.79% | 30.11% |
| 13b-linear-ft-mnist | 98.90% | 22.63% | 20.83% | 10.35% |
| 13b-linear-ft-cifar10 | 64.98% | 92.15 % | 35.29% | 31.55% |
| 13b-linear-ft-cifar100 | 39.17% | 43.07% | 86.42% | 27.03% |
| 13b-linear-ft-miniImagenet | 47.75% | 29.77% | 29.89% | 91.24% |
| 13b-lora-ft-mnist | 99.85% | 38.09% | 30.05% | 15.58% |
| 13b-lora-ft-cifar10 | 23.43% | 95.50% | 5.48% | 19.68% |
| 13b-lora-ft-cifar100 | 5.48% | 3.24% | 93.16% | 14.20% |
| 13b-lora-ft-miniImagenet | 0.24% | 4.99% | 5.22% | 95.76% |

Table 9: EMT evaluation accuracy of 3-epoch linear/lora fine-tuned `LLaVA-7/13b` on MNIST, CIFAR-10, CIFAR-100, and *mini*Imagenet, against `LLaVA-7/13b`.

| Checkpoints | Datasets | | | |
|---|---|---|---|---|
| | MNIST | CIFAR-10 | CIFAR-100 | *mini*Imagenet |
| 7b-v0 | 56.96% | 56.71% | 34.53% | 14.12% |
| ft-mnist-1ep | 97.21% | 9.20% | 10.38% | 5.88% |
| ft-cifar10-1ep | 20.45% | 90.54% | 47.34% | 34.90% |
| ft-cifar100-1ep | 26.16% | 64.54% | 72.94% | 33.86% |
| ft-miniImagenet-1ep | 0.67% | 49.46% | 38.07% | 68.69% |
| ft-mnist-2ep | 98.31% | 9.27% | 9.92% | 5.91% |
| ft-cifar10-2ep | 17.96% | 91.63% | 40.76% | 32.36% |
| ft-cifar100-2ep | 28.21% | 52.26% | 76.68% | 30.86% |
| ft-miniImagenet-2ep | 2.10% | 44.95% | 31.37% | 75.48% |

Table 10: Finetuning `LLaVA-7b-linear` under different epochs.

| Checkpoints | Datasets | | | |
|---|---|---|---|---|
| | MNIST | CIFAR-10 | CIFAR-100 | *mini*Imagenet |
| 7b-v0 | 56.96% | 56.71% | 34.53% | 14.12% |
| ft-mnist-1ep | 98.52% | 11.41% | 0.57% | 0.28% |
| ft-cifar10-1ep | 0.01% | 89.92% | 0.44% | 15.58% |
| ft-cifar100-1ep | 0.01% | 3.29% | 84.46% | 13.17% |
| ft-miniImagenet-1ep | 0.01% | 8.04% | 4.78% | 88.77% |
| ft-mnist-2ep | 99.24% | 13.72% | 1.18% | 0.86% |
| ft-cifar10-2ep | 0.01% | 91.89% | 1.39% | 15.96% |
| ft-cifar100-2ep | 0.01% | 2.55% | 88.00% | 13.59% |
| ft-miniImagenet-2ep | 0.00% | 5.77% | 5.61% | 92.03% |

Table 11: Finetuning `LLaVA-7b-lora` under different epochs.

