# OpenReview forum: "Investigating the Catastrophic Forgetting in Multimodal Large Language Model Fine-Tuning"
_CPAL.cc/2024/Conference — CPAL 2024 (Proceedings Track) Oral_

### Official Review · Reviewer_DnDM · 2023-10-07
**Investigation of catastrophic forgetting in MLLMs**

**Rating:** 6
**Confidence:** 2

**Review:**

This paper introduces the EMT framework, an approach designed to assess the phenomenon of catastrophic forgetting resulting from the fine-tuning of multimodal large language models (MLLMs). The underlying concept involves treating MLLMs as image classifiers and subjecting them to varying degrees of fine-tuning.
The central finding of this study is the observation that moderate fine-tuning on one dataset confers benefits to datasets not utilized during the fine-tuning process, while excessive fine-tuning leads to catastrophic forgetting. This conclusion aligns seamlessly with logical expectations. When one heavily fine-tunes a model on a specific dataset, the anticipated outcome is an exceptionally high level of accuracy on that specific dataset. There exists no inherent mechanism within the fine-tuning process to incentivize the model to retain optimal performance on non-fine-tuned datasets.
In essence, it begs the question: Why should we anticipate any different outcome?

A few comments and questions for authors:
1. Regarding Section 4.2, titled "Reasons for the Performance Degradation," it appears that this section delves into an investigation and speculation of the factors contributing to failed predictions. However, it lacks evidence establishing a causal link between these identified reasons and the observed decline in model performance. To enhance clarity, I suggest renaming this section to something like "Analyzing Failure Modes of MLLMs."
2. In my humble opinion, fine-tuning models with billions of parameters (e.g., 7B and 13B) on datasets such as CIFAR10 and MNIST may raise concerns about the reliability of the experimental setup.
3. Have the authors explored techniques rooted in gradual unfreezing of layers, such as ULMFiT? These techniques are specifically designed to mitigate the issue of catastrophic forgetting during fine-tuning on downstream datasets. It would be insightful to know whether these methods were considered in the experimentation process.
4. In Section 3.1, would it be advantageous to employ two distinct classification heads for subsets of datasets with non-overlapping classes? The current approach, which adjusts the weights of classes from the pre-trained dataset based on their presence in fine-tuning datasets, may inadvertently encourage the classifier head to prioritize classes currently present in the dataset. This approach lacks a mechanism for preserving features learned for classes from the pre-trained dataset.
5. Could the authors provide clarification on the meaning of "less potent than LLaMa"?
6. The authors mention the use of the OpenAI-API for evaluating results. It is worth noting that this choice may present issues related to the reproducibility of the paper's findings over time, and I would strongly suggest employing an additional (reproducible) method for evaluation purposes.
7. In Section 3.1, did the authors experiment with restarting the learning rate when transitioning from the pre-trained dataset to the fine-tuned dataset? Such a strategy can impact the model's adaptation to new data, and it would be interesting to know if it was explored in the study.

---

### Official Review · Reviewer_2R6f · 2023-10-10
**Review: Evaluating Catastrophic Forgetting in Multimodal Large Language Models**

**Rating:** 7
**Confidence:** 3

**Review:**

The paper introduces EMT, a novel method for evaluating catastrophic forgetting in multimodal large language models (MLLMs) by treating them as image classifiers, revealing that while early-stage fine-tuning of MLLMs can enhance performance across various image datasets by aligning text and language features, prolonged fine-tuning tends to induce hallucinations in the models, ultimately limiting their generalizability and indicating room for improvement in current MLLM fine-tuning methodologies.

Pros:
1. The EMT method is an innovative approach to assess catastrophic forgetting in MLLMs, providing a different lens by evaluating them as image classifiers
2. The paper provides a interesting and crucial insight of fine-tuning: early-stage fine-tuning appears to enhance performance across various image datasets, while later fine-tuning can make MLLMs  hallucinate, diminishing their generalizability even when the image encoder is frozen.
3. The paper is well-written and nicely presented. The experiments are thorough and extensive. This paper opens avenues for subsequent research.

Cons：
1. While EMT offers valuable insights, its applicability and transferability to varied multimodal tasks beyond image classification (segmentation/detection) may need further validation.
2. While the paper expertly identifies and diagnoses issues related to catastrophic forgetting and hallucination in MLLMs, it may lack in offering concrete, actionable solutions or mitigation strategies to address these identified issues.
3. The phenomenon of models beginning to "hallucinate" during fine-tuning is deeply intriguing and could be elaborated further. The paper might benefit from a more in-depth analysis, exploring why and how these hallucinations occur and the intrinsic factors within the MLLMs that contribute to this issue.

---

### Official Review · Reviewer_QqLH · 2023-10-15
**Review: This paper provides an empirical analysis of catastrophic forgetting in MLLM that is experimentally rich but lacks clear theoretical analysis/support/advancement, which may limit its broader relevance.**

**Rating:** 6
**Confidence:** 4

**Review:**

**Quality:**
The submission demonstrates a methodical approach to understanding catastrophic forgetting in MLLMs through the introduction of the EMT framework. The empirical validation is well-executed with examinations of several open-source fine-tuned MLLMs and a deeper dive into fine-tuning LLaVA.

**Clarity:** The paper is structured in a manner that elucidates the problem, the proposed solution, and the evaluation method which enhances its clarity.

**Originality:**
(a) The introduction of the EMT framework appears to be a novel contribution to the field;
(b) The paper delves into an area that seems less explored in the context of MLLMs, which adds a degree of originality to the work.

**Significance:** The insights garnered from the study, particularly around the effects of fine-tuning, are pertinent and could be instrumental in guiding future work in MLLM fine-tuning.

**Pros:**
- Rich experimental design providing a robust empirical analysis of catastrophic forgetting in MLLMs.
- Introduction of the EMT framework as a novel method to evaluate catastrophic forgetting in MLLMs.
- The paper addresses a relevant and challenging problem in the domain, which could stimulate further research.

**Cons:**
- The work may benefit from a stronger theoretical foundation to support the empirical findings.
- The significance of the contribution may not be very clear or substantial, as noted, and is heavily skewed towards an empirical study rather than novel theoretical or practical contributions.
- A more detailed exploration of the EMT framework's post-processing methods and their implications could provide a fuller understanding of the evaluation process.

---

### Meta-Review · Area_Chair_9vK2 · 2023-11-19

**Recommendation:** Accept (Poster)
**Confidence:** 4

**Metareview:**

This paper proposes a novel framework, EMT, to evaluate the catastrophic forgetting phenomenon in multimodal large language models (MLLMs) by treating each of them as an image classifier. Interestingly, they demonstrate that despite early improvements in aligning text and image features, extended fine-tuning results in reduced generalizability and performance issues, highlighting the need for improvement in MLLM development and fine-tuning methods.

Most reviewers agree that EMT is innovative and the observation of models beginning to "hallucinate" during fine-tuning is intriguing.  Although some reviewers raise concerns on the limitations of EMT, e.g., only constraints to classification, limited contributions to theoretical analysis or approaches for catastrophic forgetting, rediscovering and evaluating a classical phenomenon in modern multimodal finetuning setting is still widely interesting to CPAL audience and community. Therefore, I recommend the acceptance of this paper.

---

### Decision · Program_Chairs · 2023-11-19

**Decision:**

Accept (Oral)

**Comment:**

The paper introduces the EMT framework for evaluating catastrophic forgetting in multimodal large language models (MLLMs). It treats MLLMs as image classifiers and shows that early-stage fine-tuning enhances performance across image datasets, but prolonged fine-tuning induces hallucinations, limiting generalizability. Reviewers appreciate the innovative EMT method and the insight into fine-tuning but suggest further exploration of its applicability to diverse multimodal tasks, providing concrete solutions for the identified issues, and a more in-depth analysis of why hallucinations occur. Additionally, addressing reproducibility concerns and experimenting with learning rate strategies during transitions are suggested. Overall, the paper offers valuable contributions to understanding catastrophic forgetting in MLLMs but could benefit from more depth in understanding why and how these hallucinations occur.

The action PC chair for this paper is Atlas Wang, who made the decision after carefully reading the paper as well as the comments by all reviewers and AC. The decision is agreed by all PC chairs.